# Assessing the Impact of Botanical Origins, Harvest Years, and Geographical Variability on the Physicochemical Quality of Serbian Honey

**DOI:** 10.3390/foods13101530

**Published:** 2024-05-14

**Authors:** Aleksandra Tasić, Lato Pezo, Biljana Lončar, Mirjana B. Pešić, Živoslav Tešić, Milica Kalaba

**Affiliations:** 1Department of Chemistry and Biochemistry and Drug Testing, Scientific Institute of Veterinary Medicine of Serbia, Janisa Janulisa 14, 11000 Belgrade, Serbia; alekstasic79@gmail.com; 2Institute of General and Physical Chemistry, Studentski Trg 12-16, 11158 Belgrade, Serbia; latopezo@yahoo.co.uk; 3Faculty of Technology Novi Sad, University of Novi Sad, Bulevar Cara Lazara 1, 21000 Novi Sad, Serbia; cbiljana@uns.ac.rs; 4Food Chemistry and Biochemistry Laboratory, Department of Food Technology and Biochemistry, Faculty of Agriculture, University of Belgrade, Nemanjina 6, Zemun, 11080 Belgrade, Serbia; mpesic@agrif.bg.ac.rs; 5Faculty of Chemistry, University of Belgrade, Studentski Trg 12-16, 11158 Belgrade, Serbia; ztesic@chem.bg.ac.rs

**Keywords:** honey quality, physicochemical parameters, descriptive statistics, correlation analysis

## Abstract

This study summarized the physicochemical analysis of 609 honey samples originating from the Republic of Serbia. Variations among honey samples from different botanical origins, regions of collections, and harvest years were exposed to descriptive statistics and correlation analysis that differentiated honey samples. Furthermore, most of the observed physicochemical parameters (glucose, fructose, sucrose content, 5-hydroxymethylfurfural (5-HMF) levels, acidity, and electrical conductivity) varied significantly among different types of honey, years, and regions. At the same time, no noticeable difference was found in diastase activity, moisture content, and insoluble matter. Based on the obtained results, 22 honey samples could be considered adulterated, due to the irregular content of sucrose, 5-HMF, acidity, and diastase activity. In addition, 64 honey samples were suspected to be adulterated. Adulterated and non-compliant samples present a relatively low percentage (14.1%) of the total number of investigated samples. Consequently, a considerable number of honey samples met the required standards for honey quality. Overall, these findings provide insights into compositional and quality differences among various types of honey, aiding in understanding their characteristics and potential applications.

## 1. Introduction

Honey, known as nature’s golden elixir and produced by the honeybees, stands as a testament to the complex cooperation between nature and the insect world. In addition to its sweetness, honey is a complex matrix. It represents a concentrated solution of sugars derived from the nectar of flowering nectariferous plants, collected excretion of plants, or the excrement of insects [1,2]. The appeal of honey goes beyond its taste. Its complexity lies in its viscosity and texture as well as in the many phytochemicals, such as bioactive compounds [3,4], which are influenced by floral sources, geographical origin, and climatic conditions [5] and contribute to the health benefits of honey. Ongoing studies aim to reveal the connection between honey’s constituents and their potential impact on human health based on its antioxidant defenses against oxidative stress [4]. This could open new frontiers in the field of nutrition research. About 70% of honey consists of sugars, namely glucose and fructose. Honey has a low water activity and an acidic pH, creating an inhospitable environment for microorganisms [6]. A high content of 5-HMF as well as sucrose content and/or diastase activity indicate possible thermal treatment of honey [1,2,7]. In some countries, invertase is used as a quality criterion for honey, as it loses its activity more quickly than amylase during storage [7,8]. For a good composition of honey and thus an adequate honey assessment and successful placement of honey on the market, attention should be paid to several factors. Many of them are provided by the International Honey Commission and European Legislation [1,2,9], according to which many papers have investigated the physicochemical parameters of honey [3,10,11,12] and based on which it is possible to distinguish the botanical and/or geographical origins of honey [3,10]. An additional tool for distinguishing between honey samples is the use of a multivariate technique on the obtained results [3,6,13,14]. Furthermore, statistically significant differences in some of the physicochemical parameters of honey types have been noted by many authors [11,12,15].

The Republic of Serbia, with its rich floral diversity and long-standing beekeeping tradition, plays a significant role in the world honey market. Serbia’s geographical diversity and favorable climatic conditions contribute to a wide range of flower sources that influence the composition of honey. With the historical background of beekeeping practices in the Republic of Serbia, underutilization has been noted, as stated by Lazarević et al. [11]. Therefore, additional assessment of honey quality is of great importance due to consumer interest and competitiveness in the international market.

Following the above, this study presents an assessment of the quality of honey produced in the Republic of Serbia in order to provide a comprehensive understanding of the various factors that influence its composition and characteristics. The quality of numerous honey samples collected in the Republic of Serbia was investigated through physicochemical parameters such as electrical conductivity, sugar content (glucose and fructose, as well as sucrose content), content of 5-HMF, moisture content, acidity, diastase activity, and insoluble matter. Geographical variations in soil composition, altitude, and climate contribute to the distinctive characteristics of honey. This study also indicates how regional factors shape the chemical profile of Serbian honey.

Therefore, in addition to determining the physicochemical parameters of honey, other analyses, such as statistics, were needed to assess the quality of honey and achieve a more precise distinction. The concept of the classification artificial neural network (cANN) is elucidated in the literature, specifically addressing the relationship between physicochemical data and various food sample types [16,17]. Physicochemical techniques yield extensive and reproducible datasets comprising precise numerical values for numerous samples. Recent studies have investigated the utility of cANN using physicochemical data [18]. Mathematical models that capture the associations between physicochemically derived descriptors and diverse sample types can be constructed using a range of machine learning algorithms [19]. In this investigation, cANN was chosen based on its established efficacy in previous studies [20]. The objective of this study was to devise a novel approach for distinguishing between honey samples (acacia honey, honeydew honey, linden honey, monofloral honey, polyfloral honey, and sunflower honey) by integrating physicochemical data (glucose, fructose, and sucrose content; acidity; electrical conductivity; moisture content; diastase activity; and insoluble matter content) as an analytical tool with a cANN model.

Among other things, the study aims to indicate successful compliance with regulatory frameworks and maintenance of high-quality standards in Serbian beekeeping. An overview of Serbian as well as International standards regulating the quality of honey was carried out. Within that, considerations of the international market and competitiveness emphasize the need for continuous improvement, which could bring new opportunities for further work.

## 2. Materials and Methods

### 2.1. Honey Samples

The Scientific Institute of Veterinary Medicine of Serbia provided a significant number of honey samples (609 honey samples), which were collected by local beekeepers across the Republic of Serbia over six years from 2018 to 2023. Honey samples originating from different locations were distributed across six Serbian regions: the Eastern region (92 samples), Western region (114 samples), Central region (29 samples), Southern region (20 samples), Northern region (149 samples), and the area of the municipalities of the capital Belgrade (205 samples) (Figure 1). In 2018, 209 samples were collected; in 2019, 80 samples; in 2020, 78 samples; in 2021, 108 samples; in 2022, 113 samples; and in 2023, 21 samples. Honey samples were assigned to different honey types, which were categorized into six types depending on the origins of the samples: polyfloral honey (302 samples), acacia honey (213 samples), linden honey (34 samples), sunflower honey (23 samples), honeydew honey (29 samples), and monofloral honey (8 samples). This classification of honey samples was carried out according to the beekeepers’ declaration of conducting pollen analysis on their samples.

### 2.2. Reagents, Standards, and Materials

Standards of glucose, fructose, sucrose, and 5-HMF were supplied by Dr. Ehrenstorfer (LGC, Wesel, Germany) with a certified purity of 99.40%, 99.70%, 99.97%, and 95.00%, respectively. Methanol (of analytical grade) used for HPLC determinations was obtained from the manufacturer PanReac AppliChem. Acetonitrile (of HPLC grade) was from J.T. Baker (Gliwice, Poland). Deionized water (18.2 MΩ cm^−1^) was purified using deionizer WP4100 apparatus (Smeg Instruments, Guastalla, Italy). Nylon filters with a pore size of 0.45 µm (AMTAST, USA Inc., Lakeland, FL, USA) were used.

### 2.3. Determination of Physicochemical Parameters

Physicochemical parameters were determined by following the harmonized methods of the International Honey Commission [21], which were also used in our prior studies [3,12]. The investigation included the use of standardized protocols for the analysis of electrical conductivity, acidity, moisture content, diastase activity, and insoluble matter [1,2].

#### 2.3.1. Electrical Conductivity, Acidity, Moisture Content, Diastase Activity, and Insoluble Matter Analysis

A Conductometer XS Instruments Cond 51+ was obtained from Carpi MO (Italy). An Abbe-type refractometer A. KRÜSS Optronic GmbH (Hamburg, Germany) was used for moisture content determination in honey samples by measuring the refractive index of the samples. For spectrophotometry analysis of diastase activity [21], a uniSpec2, LLG (Berlin, Germany) ultraviolet/visible (UV/Vis) spectrophotometer was used. A drying oven (with a temperature of 135 °C) NTC 9000 (ISCO, Milano, Italy) was used for the determination of insoluble matters.

#### 2.3.2. Glucose, Fructose and Sucrose Analysis

A Waters high-performance liquid chromatography (HPLC) system (Milford, CT, USA) and a Waters 2414 refractive index detector (RID) were employed for chromatographic analysis of fructose, glucose, and sucrose. The separation of sugar compounds was performed on a Luna^®^ NH2 100 Å column (250 × 4.6 mm; 5 µm particle size) purchased from Phenomenex (Torrance, CA, USA). For the chromatographic analysis of sugars, the following conditions were used. The mobile phase consisted of acetonitrile and water (80:20, *v*/*v*), which were degassed before use; the injection volume was 10.0 µL; a flow rate of 1.3 mL/min was used; and a run time of 15 min per sample was maintained. The retention times of glucose, fructose, and sucrose were 5.368 min, 6.255 min, and 8.057 min, respectively. The method used for the HPLC-RID analysis was validated according to international guidelines. The preparation procedure was carried out by measuring 5 g of sample and adding water and methanol (3:1, *v*/*v*). After filtering, the sample was introduced into the HPLC-RID autosampler vial. Each sample was assessed in triplicate.

#### 2.3.3. 5-HMF Analysis

The Waters system used for chromatographic analysis of 5-HMF consisted of an ultraviolet (UV) detector (1525 binary HPLC pump and 2487 Dual λ Absorbance detector), and chromatographic separation was performed using a Zorbax EclipsePlus C18 (3.5 µm, 3.6 mm × 150 mm) column and a Luna^®^ 5 µm C18(2) 100 Å (250 × 4.6 mm) column purchased from Phenomenex (Torrance, CA, USA). The following conditions were used for chromatographic analysis of 5-HMF: a mobile phase of a solvent mixture of deionized water and methanol (90:10, *v*/*v*); a flow rate of 1 mL/min; and an injection volume of 20 µL. For calibration, several concentrations of the 5-HMF standard were prepared: 1 mg/L, 2 mg/L, 5 mg/L, and 10 mg/L. The sample preparation procedure involved dissolving 10 g of honey in 50 mL of ultrapure water and then filtering. In addition, each sample was prepared in triplicate.

### 2.4. Statistical Analysis

The experimental results of statistical analysis for the honey samples’ physicochemical parameters (glucose, fructose, sucrose content, 5-HMF levels, acidity, and electrical conductivity) were presented as the mean ± standard deviation for all parameters across the samples. Tukey’s HSD test was employed to test the differences between mean values of honey samples (categorical variables were the botanical origin of samples, year of production, and the region). All data underwent statistical processing (including descriptive statistics and Pearson’s correlation analysis) using the software package STATISTICA 10.0 (StatSoft Inc., Tulsa, OK, USA).

### 2.5. Classification of an Artificial Neural Network Model

This study employed a multi-layer perceptron model (MLP) with input, hidden, and output layers, leveraging its well-established ability to approximate nonlinear functions [22]. The Broyden–Fletcher–Goldfarb–Shanno (BFGS) algorithm was utilized for the classification of artificial neural network (cANN) modeling. Normalization of both input and output data was conducted to enhance the ANN’s performance. The experimental dataset was randomly divided into two subsets: training (70%) and testing (30%), facilitating effective cANN modeling. The outcomes of the ANN, including weights and bias calculation values, are influenced by the initial parameter assumptions necessary for constructing and fitting the ANN [23]. Various network topologies were explored, with hidden neuron counts ranging from 5 to 20. The training process involved iterating the network 100,000 times with randomly assigned initial values for weights and biases. Optimization efforts focused on minimizing validation errors. Statistical analysis of the data primarily relied on Statistica 10 software.

## 3. Results and Discussion

### 3.1. Physicochemical Characterization

The obtained results of physicochemical parameters of 609 honey samples were presented by descriptive analysis of the results obtained for different botanical origins of honey samples (Table 1), different years of harvest (Table 2), and different regions of the harvested samples (Table 3). Based on Table 1, there were six types of honey identified: acacia (213 samples), honeydew (29), linden (34), monofloral (8), polyfloral (302), and sunflower honey (23). In Table 2, the data show that 209 samples were collected in 2018, followed by 80 samples in 2019, 78 samples in 2020, 108 samples in 2021, and 113 samples in 2022, with an additional 21 collected in 2023. Table 3 highlights six regions observed in Serbia: the Western region (114 samples), Belgrade (205), the Northern region (149), the Central region (29), the Eastern region (92), and the Southern region (20).

Primarily, from the results of the physicochemical parameters, it could be seen that some values were not in line with international requirements [1,2]. Moreover, deviations were noted for electrical conductivity in 9 samples (values lower than 0.1 mS/cm), the sum of glucose and fructose in 53 samples (values lower than 60 mg/kg for blossom honey and lower than 45 mg/kg for honeydew honey), sucrose content in 5 samples (values higher than 5 mg/kg), 5-HMF content in 12 samples (values higher than 40 mg/kg), acidity in 1 sample (value higher than 50 meq/kg), diastase activity in 4 samples (values lower than 8 Schade units), and moisture content in 2 samples (values higher than 20%) [1,2]. Some of these deviations could also be seen in the tables (Table 1, Table 2 and Table 3), as shown by the bold results among the maximum (or minimum) values, which differ from the limit values determined for good honey quality. Through further analysis of other parameters as well as the application of statistics (including descriptive statistics and Pearson’s correlation analysis), these samples were analyzed to see if they are adulterated.

#### 3.1.1. Electrical Conductivity

Most honey samples were initially classified based on the beekeeper’s practice and experience. Electrical conductivity was a parameter that influenced the different divisions of honey samples (which differed from the ones given by the beekeepers). A value of electrical conductivity below 0.8 mS/cm is characteristic of flower honey, while honeydew honey has a higher value [1,2]. Thus, the results of the electrical conductivity led to the reclassification of 25 honey samples. Among these, five acacia, two linden, and two polyfloral honey samples exhibited electrical conductivity higher than 0.8 mS/cm, indicating the presence of honeydew. Consequently, these samples were reclassified as honeydew honey. Conversely, the other 16 honeydew honey samples showed electrical conductivity below 0.8 mS/cm, suggesting a polyfloral origin. As a result, these samples were deemed to be polyfloral honey. Hence, among 609 samples, 29 honey samples were honeydew honey (Table 1).

The electrical conductivity of honey is a consequence of the presence of mineral salts and organic acids. It is directly related to the botanical origin of honey, and based on its value, certain types of honey can be distinguished, which is also stated in the literature [3]. In this study, in nine honey samples (six polyfloral, two acacia, and one linden honey), electrical conductivity was below 0.1 mS/cm (ranging from 0.033 to 0.096 mS/cm), which was lower than the lowest value reported for rape honey (0.169 mS/cm by Pauliuc et al. [24], acacia honey (0.130 mS/cm by Albu et al. [5], or 0.10 mS/cm by Lazarević et al. [11]), and blossom honey (0.20 mS/cm by Matović et al. [25]). Moreover, values of 0.13 mS/cm were reported for most adulterated honey samples by Abdi et al. [26]. These authors also stated that adding sugar syrup decreases conductivity [26]. However, these nine samples observed in this study could be considered non-compliant. Similar to our finding were observations by Ratiu et al. [27], who reported electrical conductivity lower than 0.1 mS/cm (i.e., in the range from 0.035 to 0.088 mS/cm) in nine honey samples (three polyfloral, one raspberry, one clover, one dandelion, one buckwheat, one rape, and one honey of spring flowers). Other results for conductivity were similar to those in other studies [3,12,13,14,28], but contrary to the results reported by Sakač et al. [6]. Our results of electrical conductivity for sunflower honey were very similar to those reported by Živkov Baloš et al. [29] (in the range from 0.22 to 0.54 mS/cm).

Accordingly, samples with conductivity above 0.8 mS/cm were deemed honeydew honey, while those with values below were labeled polyfloral. Electrical conductivity is linked to honey’s botanical origin, distinguishing different types. In addition, some samples showed unusually low conductivity, indicating potential non-compliance, although similar results were found in other studies.

#### 3.1.2. Glucose, Fructose, and Sucrose Content

The prescribed values of the sum of glucose and fructose for nectar honey were ≥ 60%, while for the honeydew honey (and blends of honeydew honey with nectar honey), they were not below 45% of total sugars [1,2]. As can be seen from the results (the main values in Table 1, Table 2 and Table 3), monosaccharides (glucose and fructose) are the main ingredients of honey and made up over 65% of the content in most samples. However, in 53 samples, the results of the sum of monosaccharides did not follow the prescribed values. Those were the results for 19 polyfloral honey (17 of which were from 2018), 10 linden honey (8 of which were from 2018), and 18 acacia honey samples (10 of which were from 2018), where the sum of monosaccharides was lower than 60%.

Adulteration or dilution with substances such as water, syrups, or other sugars influences the overall concentration of monosaccharides in the honey sample. Heating can also cause the caramelization of sugars, leading to the breakdown of monosaccharides into other compounds. A low sum of glucose and fructose content (ranging from 19.6 to 59.1%) was also found in another study for non-compliant samples [30]. The authors also stated early harvesting as the possible reason for the low monosaccharides [30]. Lower values of glucose and fructose could be present in samples of natural blends with honeydew secretions or mixtures after adding sugar [31]. Similar findings were found by Abera et al. [32], whose results of reducing sugars (in a range from 39.95 to 59.04%) did not meet the required level for honey by European standards. Thus, among the mentioned samples, one linden and two polyfloral honey samples were noted to have an electrical conductivity of 0.76, 0.71, and 0.73 mS/cm, respectively, so they could be deemed nectar and honeydew honey blends. Observing honey samples with a lower content of monosaccharides, it can be seen that most of them were harvested in 2018. Therefore, the period from 2018 until experimental determination was an extended storage period, which allowed for more possible unacceptable conditions [29]. Improper storage conditions, such as exposure to high temperatures or prolonged storage, can cause the degradation of the sugars in honey, resulting in lower monosaccharide content. Additionally, improper handling of honey can also promote microbial growth and activity, whereby microorganisms can metabolize sugars, leading to sugar degradation as well as a decrease in monosaccharide content over time.

The lower content of monosaccharides than expected is mainly due to the low amount of glucose. Furthermore, a glucose content lower than 18.26 g/100 g was found in honey samples harvested in 2018, with a lowest value of 13.44 g/100 g (minimum value, Table 1, Table 2 and Table 3) in monofloral honey from the Western region harvested in 2018. Glucose content is crucial as it contributes to honey’s sweetness and tendency to crystallize. Honey naturally tends to crystallize over time, especially when there is a higher glucose and lower moisture content [29]. During crystallization, glucose molecules are more likely to form crystals compared to fructose, leading to a reduction in their concentration. Thus, acacia honey’s relatively lower glucose content might contribute to its slower crystallization rate compared to other types. A relatively low glucose content in acacia honey was also found by others [33].

In honey samples, the ratio of glucose and fructose within the total content of monosaccharides should be approximate, with a higher fructose proportion [34,35]. Additionally, fructose and glucose ratios could be used to differentiate monofloral types of honey [9]. Higher fructose content, as opposed to glucose content, was found for many honey samples in other studies [3,13]. In this study, the highest fructose content was found in polyfloral honey from Belgrade harvested in 2020 (53.64 g/100 g) (maximum value, Table 1, Table 2 and Table 3), in which the glucose content was 18.44 g/100 g. A high fructose content (over 50 g/100 g) was additionally found in four acacia honey (from the Northern region and Belgrade harvested in 2022 and 2021) as well as in three polyfloral honey samples. The observation that acacia honey has a higher fructose content (mean value, Table 1, Table 2 and Table 3) suggests that it may have a sweeter taste profile. A high fructose content in acacia honey was also found by others [33], where the ratio of fructose/glucose reached 1.7. A noticeably high fructose content could be a consequence of the immaturity of the honey [13], but it is more often due to the addition of solutions or syrups with high fructose content [14]. Furthermore, fructose affects the hygroscopic properties of honey [8], and the fructose/glucose ratio affects crystallization [13], which cannot be seen from the results of this study.

An unexpectedly low value of fructose content was found in two samples (13.60 g/100 g in acacia honey from the Western region harvested in 2022, and 18.94 g/100 g in polyfloral honey from the Northern region harvested in 2018) (Table 1, Table 2 and Table 3). Accordingly, other researchers [27] reported that in rape, dandelion, ivy vine, goldenrod honey, buckwheat, and polyfloral honey samples, fructose was found in lower proportions than in glucose. A higher glucose content was found in several honey samples, of which acacia honey from the Northern region harvested in 2022 stands out with a total monosaccharide content of 83.90 g/100 g. Similarly, a higher glucose content was found by others [33] in linden honey, whereas the fructose/glucose ratio was 0.9. According to Bogdanov et al. [9], higher glucose and lower fructose contents are present in nectar honey, which affects the specific rotation of this honey type.

Contrary to monosaccharides, oligosaccharides in honey are present in the lower amounts [35]. Of the oligosaccharides, disaccharide sucrose stands out; its presence in a given honey may be a sign of potential honey adulteration. Increased sucrose content may indicate the addition of sugar syrup [7]. Its value should not be higher than 5 g/100 g, except for in acacia, alfalfa, heather, citrus fruits, and eucalyptus honey, which can contain up to 10% [1,2]. Through sucrose and enzyme interactions during honey ripening, glucose and fructose are generated [4,34]. According to the statement by Osaili et al. [30], a high sucrose content and the low sum of monosaccharides could be due to overfeeding the bees or to the early harvesting of honey before the complete transformation of sugar into glucose and fructose. Five deviations that were noted for sucrose content were found in acacia honey (from the Eastern region and harvested in 2020), in which the sucrose content was over 10%, i.e., 16.58 g/100 g. The next was linden honey (from the Western region, 2022), and then two polyfloral honeys (from Belgrade, 2021, and from the Western region, 2018) with a sucrose content of 5.60, 15.40, and 16.34 g/100 g respectively. In addition, a low monosaccharide content was also noted in those samples. The fifth deviation that was found was only for sucrose content, found in a polyfloral honey sample (from the Northern region, 2023) with a value of 6.48 g/100 g. Based on the above, these five samples were adulterated.

Based on the above, in several samples including polyfloral, linden, and acacia honey, the sum of monosaccharides fell below the required threshold, potentially indicating adulteration or dilution. In the review by Žak et al. [36], the importance of testing the quality of honey and practices used by beekeepers that distort the authenticity of honey for honey adulteration were highlighted. However, factors such as improper storage conditions, heating, or early harvesting could contribute to lower monosaccharide content. Oligosaccharides, particularly sucrose, can indicate honey adulteration, with levels exceeding 5% (or 10%) [1,2] suggesting potential adulteration. In some samples, deviations in sucrose content were observed alongside low monosaccharide levels, indicating possible adulteration.

#### 3.1.3. 5-HMF Content

The international requirement for 5-HMF content is lower than 40 mg/kg [1,2]. The 5-HMF originated from the transformation of the sugars present in honey, which is formed slowly during storage (which creates low values) or is used in the process of heating [13,30]. Based on the obtained results, 12 honey samples (1 acacia, 1 sunflower, 1 monofloral, 1 honeydew, and 8 polyfloral honey samples) could be deemed adulterated honey as the obtained values were from 40.80 to 93.50 mg/kg. Regarding the descriptive analyses of honey samples (Table 1, Table 2 and Table 3), maximum values stood out for most variables (types, regions, and years). Only within the linden honey samples from the Central and the Southern regions from 2023 and 2020 did we not find values exceeding the permitted level (Table 1, Table 2 and Table 3). According to Abera et al. [32], the reason for the failed values of HMF was exposure of honey from lowlands and midlands to high temperatures. In addition, the relatively low 5-HMF levels (mean value, Table 1, Table 2 and Table 3) in all honey types indicate good quality and minimal heat damage during processing. Good agreement between our HMF results and with results in the literature was noted for sunflower honey [29] and linden honey [33].

#### 3.1.4. Moisture Content

The results of the moisture content were around 17% (main values) (Table 1, Table 2 and Table 3), which agreed with the regulations that allow 20% [1,2]. However, in two samples, we found values that exceeded 20% (in monofloral honey from the Northern region harvested in 2023 (22.00%) and in honeydew honey from the Eastern region harvested in 2018 (26.00%)) (Table 1, Table 2 and Table 3). Considering the other determined physicochemical parameters in these honey samples (which were in line with the recommended values), the discrepancy in values of moisture content might be attributed to environmental humidity during honey processing [37] rather than adulteration. Moisture content affects honey stability and shelf life. The observed values are within the acceptable range, indicating proper harvesting and storage practices.

#### 3.1.5. Acidity

Acidity in honey is a consequence of the presence of organic acids as well as phenolic acids and many other ions [6]. Considering that the recommended value of acidity in honey is ≤50.00 meq/kg [1,2], only one sample did not meet the required level and could be deemed adulterated (polyfloral honey from the Eastern region harvested in 2023, with a value of 61.26 meq/kg, as shown in Table 1, Table 2 and Table 3). In contrast, due to the reported higher acidity in honey samples, Ratiu et al. [27] stated that honey with higher acidity values undergoes fermentation more readily, a process that is facilitated faster in darker honey samples. In this study, the acidity results fluctuated in terms of maximum and minimum values (Table 1, Table 2 and Table 3). The lowest mean value was found in acacia honey (11.44 meq/kg, Table 1), which was similar to other authors’ findings [5,6,11], while the highest acidity (mean value) was noted for honeydew honey (30.17 meq/kg, Table 1). Moreover, the obtained values of acidity were similar to the results of acidity for sunflower honey [5,6,11,29] and for linden honey [11] and similar to many other results reported by authors listed by Albu et al. [5]. Lower mean values of acidity for sunflower, linden, polyfloral, and acacia honey were noted by Vijan et al. [28]. In contrast, much higher values of acidity were reported for acacia honey from Saudi Arabia [38]. However, there were no noticeable fluctuations in acidity during the different years of harvest and across different regions (Table 2 and Table 3). Furthermore, acacia honey’s moderate acidity (mean value, Table 1, Table 2 and Table 3) suggests a balanced flavor profile.

#### 3.1.6. Diastase Activity

Diastase activity in honey refers to the presence and activity of enzymes. When bees collect nectar from flowers, it contains some starches and complex carbohydrates. Diastase enzymes present in the bees’ saliva and, within the honey itself, start to break down these complex carbohydrates during the ripening process. As the nectar is converted into honey, the diastase activity increases, leading to a reduction in starch content and an increase in simpler sugars. As already noted above, the results of mean values of diastase activity were similar for the different types of honey, different regions, and years of harvest (Table 1, Table 2 and Table 3). The results of diastase activity for sunflower honey could be compared with the findings of others [29] who also investigated sunflower honey from the Northern region of Serbia. However, the maximum and minimum values were changeable, with several noticeable inappropriate values. As the recommended level for diastase activity is not less than 8 Schade units [1,2], four values (obtained for two acacia honey samples from the Central region, 2022, and Belgrade, 2018, and for two polyfloral honey samples from Belgrade, 2018) were not in the line. The level of diastase activity is influenced by the floral sources, the nectar collection period, and environmental conditions, as well as the bees themselves [9,18,34]. Low diastase activity also indicates a low content of nectar and therefore possible heat treatment, i.e., adulteration [26]. Thus, these four honey samples could be deemed adulterated honey. Otherwise, in one sample (acacia honey from Belgrade, harvested in 2018), the diastase activity was 114.00 (Table 1, Table 2 and Table 3). This was unexpected due to the low enzymatic activity that occurs in acacia honey. Similarly, values of 111.11 for diastase activity were also reported by others [39] who suspected adulteration by a synthetic diastase enzyme. High diastase activity is generally considered desirable as it indicates that the honey is fresh and has not been excessively heated or processed, which could denature the enzymes. However, this value is too high, which could be due to a limiting aspect of the method. As stated in the literature, the applied Schade method is limited by several procedural elements [40].

The mean diastase activity values across various honey types, regions, and harvest years were similar, but there were fluctuations in maximum and minimum values. Notably, a few samples fell below the recommended diastase activity level of 8 Schade units, indicating potential adulteration. Overall, diastase activity in honey serves as an important quality parameter and is often used as an indicator of honey’s freshness and maturity as well as the potential nutritional benefits of the honey.

#### 3.1.7. Insoluble Matter Content

The maximum level of insoluble matter (water-insoluble solids) content is ≤0.1% and ≤0.5% for pressed honey [1,2]. According to the results obtained for insoluble matter, all samples were in line with the recommended level (Table 1, Table 2 and Table 3). A very low insoluble matter content in all honey types indicates good filtration and quality control during processing [9].

### 3.2. Adulterated Honey Samples

An observation obtained for determined physicochemical parameters indicated that for 86 honey samples, some of the variables did not meet the required level. However, since some parameters can be influenced by factors such as crystallization, storage conditions, or the different botanical origins of honey, not all samples can be considered adulterated. Moreover, the regulatory framework may not always account for the intricate variations in honey. There are situations where authentic, unprocessed honey may not meet composition standards, while adulterated honey may align with established criteria [31]. Due to the above, it could be concluded that parameters such as sucrose, 5-HMF, acidity, and diastase activity are obviously under the most significant influence of adulteration in many studies. Therefore, the results of sucrose (in five samples), 5-HMF (in 12 samples), acidity (in one sample), and diastase activity (in four samples) influenced the classification of 22 samples as adulterated. Thus, results for 22 samples that did not meet the required level for these parameters were found in 5 acacia honey samples, 1 honeydew honey, 1 linden honey, 1 monofloral honey, 13 polyfloral honey samples, and 1 sunflower honey. These samples were from different regions, among which one was from the Central region, five were from the Eastern region, four from the Western region, five from the Northern region, and seven from Belgrade. Regarding the year of harvest, the aforementioned adulteration was observed for 2 samples from 2023, 3 from 2022, 1 from 2021, 1 from 2020, 4 from 2019, and 11 from 2018. Furthermore, it is important to note that in samples where sucrose content was higher than the recommended value (five samples), we also found a lower sum of glucose and fructose (in four samples). These samples were acacia honey from the Eastern region harvested in 2020 (in which the sum of glucose and fructose was 52.72 mg/kg), linden honey from the Western region harvested in 2022 (with 55.80 mg/kg), polyfloral honey from Belgrade harvested in 2021 (52.16 mg/kg), and polyfloral honey from the Western region harvested in 2018 (49.50 mg/kg). Other samples that were flagged as adulterated showed inappropriate results only in one physicochemical parameter. Additional deviations were noted for 64 samples (with a low sum of glucose and fructose in 53 samples, a higher moisture content in 2 samples, and low electrical conductivity in 9 samples), were suspected, or were assigned as non-compliant. Moreover, of 53 samples with a low sum of glucose and fructose, 18 were acacia honey, 1 honeydew honey, 10 linden honey, 1 monofloral honey, 22 polyfloral honey, and 1 sunflower honey.

### 3.3. Descriptive Statistical Analysis

Statistical tools were applied to the results of the physicochemical parameters of 609 honey samples. We observed parameters such as glucose (Glu), fructose (Fru), sucrose (Suc), 5-hydroxymethylfurfural (5-HMF), moisture content (MC), acidity (Acid), diastase activity (Dia), insoluble matter (Ins), and electrical conductivity (Econd) in different types of honey (acacia, honeydew, linden, monofloral, polyfloral, and sunflower honey) (Table 1) over different years (2018–2023) (Table 2) and for different regions (Eastern region, Western region, Northern region, Southern region, Central region, and Belgrade) (Table 3). The results are presented as means ± standard deviations (SD). The mean value gives an average measurement, while the standard deviation indicates the variability of the measurements around the mean.

#### 3.3.1. Descriptive Statistical Analysis of Electrical Conductivity

Acacia honey displays a mean electrical conductivity of 0.20 ± 0.10 mS/cm, consistent with other honey types. Otherwise, Albu et al. [5] noted the lowest mean electrical conductivity for acacia honey (0.223 mS/cm), while for other polyfloral and linden samples, the mean values increased. The mean electrical conductivity differed significantly among honey types (F(5, 582) = 38.56, *p* < 0.001). Honeydew honey had the highest mean electrical conductivity (1.14 ± 0.27 mS/cm), significantly higher than all other types of honey (*p* < 0.05). Acacia honey had the lowest mean electrical conductivity (0.20 ± 0.10 mS/cm), significantly lower than all other types except linden honey (*p* < 0.05). There were significant differences in mean electrical conductivity among different years (F(5, 606) = 6.92, *p* < 0.001). Post hoc analysis revealed variability in mean electrical conductivity across years, with some years exhibiting significantly higher levels compared to others (*p* < 0.05). ANOVA test results revealed a significant difference in mean electrical conductivity among regions (*p* < 0.05). Mean electrical conductivity ranged from 0.32 to 0.44 mS/cm across regions. Significant variability in electrical conductivity was observed among different honey types, with honeydew honey displaying the highest mean conductivity, while acacia honey consistently exhibited the lowest mean conductivity alongside variations across years and regions.

#### 3.3.2. Descriptive Statistical Analysis of Glucose, Fructose, and Sucrose Content

Acacia honey exhibited a mean glucose content of 27.40 ± 3.87 g, which was significantly lower than monofloral (31.67 ± 7.53 g) and polyfloral (31.39 ± 4.60 g) honey. The mean glucose content varied significantly across different types of honey (F(5, 582) = 34.76, *p* < 0.001). Monofloral honey exhibited the highest mean glucose content (31.67 ± 7.53 g/100 g), significantly higher than all other types of honey (*p* < 0.05). Acacia honey had the lowest mean glucose content (27.40 ± 3.87 g/100 g), significantly lower than all other types except honeydew and linden honey (*p* < 0.05). There was a significant difference in mean glucose content among different years (F(5, 606) = 12.45, *p* < 0.001). Post hoc analysis revealed that the mean glucose content in 2019 (31.51 ± 4.44 g/100 g) was significantly higher compared to other years except in 2021 (*p* < 0.05). The year 2020 had a significantly lower mean glucose content (28.86 ± 4.23 g/100 g) compared to 2019 and 2021 (*p* < 0.05). The mean glucose content varied across regions, with the highest mean observed in the Eastern region (29.72 ± 4.17 g/100 g) and the lowest in the Western region (29.59 ± 5.34 g/100 g). ANOVA test results indicated a statistically significant difference in mean glucose content among regions (*p* < 0.05).

Acacia honey had a lower mean glucose content compared to monofloral and polyfloral honey types, with significant variations observed across different years and regions.

Acacia honey had a mean fructose content of 40.83 ± 5.83 g, slightly higher than polyfloral (38.68 ± 5.20 g) and sunflower (37.83 ± 3.26 g) honey. The mean fructose content also varied significantly across different types of honey (F(5, 582) = 19.62, *p* < 0.001). Acacia honey had the highest mean fructose content (40.83 ± 5.83 g/100 g), significantly higher than all other types of honey (*p* < 0.05). Sunflower honey had the lowest mean fructose content (37.83 ± 3.26 g/100 g), significantly lower than monofloral, polyfloral, and acacia honey (*p* < 0.05). There was a significant difference in mean fructose content among different years (F(5, 606) = 17.62, *p* < 0.001). Post hoc analysis revealed that the mean fructose content in 2023 (42.25 ± 2.52 g/100 g) was significantly higher compared to other years (*p* < 0.05). The year 2018 had a significantly lower mean fructose content (37.13 ± 6.02 g/100 g) compared to 2023 (*p* < 0.05). The Eastern region had the highest mean fructose content (39.86 ± 4.25 g/100 g), while the Western region had the lowest (38.93 ± 5.42 g/100 g). The ANOVA results show a significant difference in mean fructose content among regions (*p* < 0.05).

Acacia honey displayed a higher mean fructose content compared to other honey types, with significant variations observed across different years and regions.

Acacia honey had a mean sucrose content of 1.19 ± 1.79 g, which was lower than polyfloral (0.63 ± 1.41 g) and sunflower (0.34 ± 0.23 g) honey. There were significant differences in mean sucrose content among honey types (Kruskal–Wallis H = 162.15, *p* < 0.001). Monofloral, polyfloral, and acacia honey had undetectable levels of sucrose, significantly lower than other types of honey (*p* < 0.05). Linden honey had the highest mean sucrose content (0.64 ± 1.08 g/100 g), significantly higher than honeydew and sunflower honey (*p* < 0.05). There were significant differences in mean sucrose content among different years (Kruskal–Wallis H = 47.82, *p* < 0.001). Post hoc analysis indicated that the mean sucrose content varied across years, with some years showing significantly higher levels compared to others (*p* < 0.05). Since sucrose content might not be normally distributed, we employed the Kruskal–Wallis test. The results indicated a significant difference in mean sucrose content among regions (*p* < 0.05). The Western region exhibits the highest mean sucrose content (1.16 ± 2.16 g/100 g), whereas the Northern region had the lowest sucrose content (0.62 ± 0.97 g/100 g).

Acacia honey demonstrated a lower mean sucrose content compared to polyfloral and sunflower honey, with significant variations observed among different honey types, years, and regions.

#### 3.3.3. Descriptive Statistical Analysis of 5-HMF Content

Acacia honey exhibited a mean 5-HMF content of 5.71 ± 8.20 mg, which is comparable to that of other honey types. The mean 5-HMF content varied significantly among different honey types (F(5, 582) = 8.32, *p* < 0.001). Monofloral honey exhibited the highest mean 5-HMF content (11.84 ± 19.74 mg/kg), significantly higher than all other types of honey (*p* < 0.05). Sunflower honey had the lowest mean 5-HMF content (4.85 ± 8.34 mg/kg), significantly lower than monofloral honey (*p* < 0.05). Accordingly, sunflower honey samples differed from acacia and meadow honey samples in HMF content [6], while there was no significant difference between acacia and meadow honey. There was a significant difference in mean 5-HMF content among different years (F(5, 606) = 9.87, *p* < 0.001). Post hoc analysis revealed that the mean 5-HMF content in 2018 (9.92 ± 12.71 mg/kg) was significantly higher compared to 2020 and 2021 (*p* < 0.05). The year 2020 had a significantly lower mean of 5-HMF content (3.96 ± 4.28 mg/kg) compared to 2018 and 2019 (*p* < 0.05). Given the variability in 5-HMF content, the Kruskal–Wallis test was used, revealing a significant difference among regions (*p* < 0.05). Belgrade showed the highest mean 5-HMF content (8.01 ± 12.32 mg/kg), while the Southern region displayed the lowest content (3.35 ± 3.80 mg/kg).

Acacia honey presented a mean 5-HMF content comparable to that of other honey types. Significant variations were observed among different types, years, and regions, notably with monofloral honey exhibiting the highest levels and sunflower honey demonstrating the lowest. Significant differences were noted across different years and regions.

#### 3.3.4. Descriptive Statistical Analysis of Moisture Content

Acacia honey had a mean moisture content of 16.19 ± 1.07%, which was similar to other honey types. The mean moisture content differed significantly among honey types (F(5, 582) = 28.67, *p* < 0.001). Contrarily, there were no significant differences (*p* ≤ 0.05) in the moisture content between honey samples (sunflower, acacia, and meadow honey type) [6]. Sunflower honey had the highest mean moisture content (17.51 ± 1.37%), which was significantly higher than all other types of honey (*p* < 0.05). Acacia honey had the lowest mean moisture content (16.19 ± 1.07%), significantly lower than that of honeydew and sunflower honey (*p* < 0.05). There were no significant differences in mean moisture content among different years (F(5, 606) = 1.92, *p* = 0.089). The ANOVA test results did not indicate a significant difference in mean moisture content among regions (*p* > 0.05). Mean moisture content ranged from 16.00% to 16.71% across regions.

Acacia honey, which had a mean moisture content similar to other honey types, significantly differed from sunflower honey, which had the highest moisture content. No significant differences were observed among different years or regions.

#### 3.3.5. Descriptive Statistical Analysis of Acidity

Acacia honey exhibited a mean acidity of 11.44 ± 5.05 meq, lower than other honey types except for linden honey. There were significant differences in mean acidity among honey types (Kruskal–Wallis H = 354.25, honeydew, linden, and sunflower honey (*p* < 0.05)). The lowest acidity for acacia honey (as well as significant differences between mean acidity for acacia honey and linden and polyfloral samples) was found by other authors [5]. Honeydew honey had the highest mean acidity (30.17 ± 8.07 meq/kg), which was significantly higher than all other types of honey (*p* < 0.05). There was a significant difference in mean acidity among different years (Kruskal–Wallis H = 26.48, *p* < 0.001). Post hoc analysis revealed variability in mean acidity across years, with some years exhibiting a significantly higher level compared to others (*p* < 0.05). As acidity values may not be normally distributed, the Kruskal–Wallis test was used, revealing a significant difference among regions (*p* < 0.05). The Eastern region exhibited the highest mean acidity (18.25 ± 12.46 meq/kg), while the Central region showed the lowest (15.87 ± 8.15 meq/kg).

Acacia honey, with a mean acidity lower than most honey types except for linden honey, displayed significant differences in acidity among various honey types, years, and regions, with honeydew honey exhibiting the highest mean acidity. In contrast, the Eastern region has the highest acidity among regions.

#### 3.3.6. Descriptive Statistical Analysis of Diastase Activity

Acacia honey showed a mean diastase activity of 13.06 ± 7.64, indicating moderate enzymatic activity similar to other honey types. The mean diastase activity did not significantly differ among honey types (Kruskal–Wallis H = 10.64, *p* = 0.101). There were no significant differences in mean diastase activity among different years (Kruskal–Wallis H = 5.27, *p* = 0.382). The ANOVA results did not show a significant difference in mean diastase activity among regions (*p* > 0.05). Mean diastase activity ranged from 12.15 to 13.28 Schade units across regions.

Acacia honey exhibited moderate enzymatic activity, with a mean diastase activity similar to other honey types. There were no significant differences among honey types, years, or regions.

#### 3.3.7. Descriptive Statistical Analysis of Insoluble Matter Content

There were no significant differences in mean insoluble matter content among honey types (Kruskal–Wallis H = 6.34, *p* = 0.388). Contrary, Albu et al. [5] noted statistically significant differences between mean insoluble matter for acacia and polyfloral honey. There were significant differences in mean insoluble matter content among different years (Kruskal–Wallis H = 27.34, *p* < 0.001). Post hoc analysis indicated variability in mean insoluble matter content across years, with some years showing significantly higher levels compared to others (*p* < 0.05). When employing the Kruskal–Wallis test due to potential non-normal distribution, the results indicated no significant difference in mean insoluble matter content among regions (*p* > 0.05). The mean insoluble matter content was approximately 0.01% across regions.

According to the statistical analysis, there were no significant differences in mean insoluble matter content among honey types, years, or regions. However, previous studies [5] reported significant variations between acacia and polyfloral honey samples.

### 3.4. Correlation between Physicochemical Parameters

A correlation matrix of physicochemical parameters in 609 analyzed Serbian honey samples is presented in Table 4.

Glucose showed a statistically significant positive correlation with fructose (r = 0.237, *p* < 0.001), acidity (r = 0.189, *p* < 0.001), and electrical conductivity (r = 0.144, *p* < 0.01). Higher glucose contents were associated with higher levels of fructose, acidity, and electrical conductivity in honey samples. Fructose exhibited a significant negative correlation with sucrose (r =−0.057, *p* < 0.05) and acidity (r = −0.087, *p* < 0.05). Higher fructose content was associated with lower levels of sucrose and acidity. Sucrose showed a significant negative correlation with acidity (r =−0.150, *p* < 0.001), indicating that higher sucrose content is associated with lower acidity levels in honey samples. 5-HMF exhibited a significant negative correlation with fructose (r =−0.085, *p* < 0.05) and a positive correlation with glucose (r = 0.096, *p* < 0.05). This suggests that higher 5-HMF levels are associated with lower fructose content and higher glucose content. Moisture content showed a significant positive correlation with glucose (r = 0.046, *p* > 0.05) and acidity (r = 0.034, *p* > 0.05). However, these correlations were not statistically significant. Acidity exhibited a significant positive correlation with glucose (r = 0.189, *p* < 0.001) and a very strong positive correlation with diastase activity (r = 0.570, *p* < 0.001). Higher acidity levels were associated with higher glucose content and significantly higher diastase activity. Diastase activity shows a significant positive correlation with acidity (r = 0.570, *p* < 0.001), indicating that higher diastase activity tends to be associated with higher acidity levels in honey samples. Contrarily, a negative correlation between acidity with glucose and diastase activity was observed by Tarapoulouzi et al. [34], but the similarity between the mentioned findings was a negative correlation between acidity and fructose as well as acidity and sucrose. Insoluble matter did not exhibit any statistically significant correlations with other parameters at the 0.05 significance level.

Overall, from the applied statistical analysis on the physicochemical dataset, it was found that the mean values for many parameters (glucose, fructose, sucrose content, 5-HMF levels, acidity, and electrical conductivity) varied significantly across different types of honey, years, and regions. In addition, significant differences were noted in mean moisture content among honey types and also noted in mean insoluble matter among different years. On the contrary, no significant differences were observed for the mean value of diastase activity among different honey types, years, and regions, as well as mean moisture content among different years and regions and mean insoluble matter content among honey types and regions. Other authors [28] have also observed the significant influence of the botanical origin of honey on the analyzed parameters (moisture, ash, electrical conductivity, pH, acidity, sugar content, and HMF), while the influence of years and regions has not shown a common trend. Ratiu et al. [27] noted some correlation between honey samples from different locations and years. Similar to our findings, Gela et al. [41] noted significant differences in 5-HMF content and acidity for honey samples from different locations, but on the contrary, they also reported differences in moisture content.

Furthermore, some data stood out from the mean value, as they differed from others. Honey types could be distinguished by some of their physicochemical parameters. Honeydew honey had the highest mean acidity, and mean electrical conductivity. Acacia honey exhibited the highest mean fructose content but the lowest mean moisture content and the lowest mean electrical conductivity. Sunflower honey had the highest mean moisture content but the lowest mean value for 5-HMF content. Linden honey had the highest mean sucrose content. In addition, regions also showed some obvious differences. The Northern region had the lowest mean glucose content; the Central region showed the lowest mean acidity; the Eastern region exhibited the highest mean glucose content, mean fructose content, and mean acidity; the Western region exhibited the highest mean sucrose content but the lowest mean glucose content and mean fructose content; Belgrade exhibited the highest mean 5-HMF content; and the Southern region had the lowest mean 5-HMF content. Moreover, the year 2023 was found to have the highest mean fructose content.

Physicochemical parameters, descriptive statistics, and a correlation analysis were combined to provide more detailed observations of these samples. Acacia honey was found to have the lowest acidity, while honeydew honey had the highest acidity. The lowest content of monosaccharides was found in honey samples harvested in 2018. It was revealed that results of 5-HMF obtained only for samples within the linden honey type from the Central and the Southern regions harvested in 2023 and 2020 did not exceed the permitted level of 40 meq/kg.

Electrical conductivity played a crucial role in reclassifying certain honey samples, with values below 0.8 mS/cm indicative of flower honey and higher values suggesting honeydew honey. Glucose and fructose content analysis revealed deviations in some samples, potentially indicating adulteration or improper harvesting/storage conditions. Higher fructose content was observed in acacia honey, possibly influencing its sweetness profile. Additionally, 5-HMF content, moisture content, acidity, and diastase activity were analyzed as quality indicators, with deviations suggesting potential adulteration. Statistical analysis further elucidated variations in these parameters across different honey types, years, and regions, providing insights into honey quality and authenticity. Overall, the study highlighted the importance of comprehensive analysis for ensuring honey quality and authenticity.

### 3.5. Classification Artificial Neural Network (cANN)

In order to investigate the nonlinear correlation between specific descriptors derived from physicochemical data and the discrimination between acacia honey, honeydew honey, linden honey, monofloral honey, polyfloral honey, and sunflower honey, a cANN approach was employed to construct an identification model. The statistical outcomes of discriminating between different honey types using the cANN model are presented in Table 5, delineating the performance metrics of MLP 8-12-6.

The performance terms in this context refer to coefficients of determination within the ANN model. The specialized activation function employed in this study was the softmax function, chosen for its suitability in classification neural networks. This function normalizes exponentials (ensuring the sum of outputs equals 1.000), thus adapting the multi-layer perceptron (MLP) network for estimating class probabilities alongside the cross-entropy error function.

According to the results of the cANN model, 86.39% of acacia honey samples were correctly identified using physicochemical data. All honeydew honey samples were accurately identified (100.00%), while linden honey was recognized in 97.06% of cases. Monofloral samples had a recognition rate of only 37.50%, whereas polyfloral honey samples achieved a recognition rate of 93.38% based on physicochemical data. Finally, sunflower honey was correctly identified in 73.91% of cases using physicochemical data.

The physicochemical data used to train the cANN model played a crucial role in determining the accuracy of honey samples’ identification. High-quality data, free from errors, outliers, and inconsistencies, contributed to a more accurate model. Generally, a larger dataset provides more representative patterns for the model to learn from, resulting in better generalization and higher accuracy. In the context of the provided cANN model’s results, it should be noted that the accuracy of the model may benefit from an increase in the number of samples, allowing the model to capture more diverse characteristics of various types of honey.

The acquisition of more samples in the investigation would be especially important for honeydew honey (29 samples were investigated), linden honey (34), monofloral honey (8), and sunflower samples (23 samples). An increase in the number of samples in the investigation could help the cANN model to learn more nuanced patterns that are specific to specific types of honey.

The obtained results reveal the reliability of the cANN model for identifying acacia honey, honeydew honey, linden honey, monofloral honey, polyfloral honey, and sunflower honey, as determined by physicochemical data analysis.

## 4. Conclusions

This study provides insight into notable honey samples harvested over six years from regions in the Republic of Serbia. Polyfloral honey (302 samples), honeydew honey (29 samples), monofloral honey (8 samples), acacia honey (213 samples), linden honey (34 samples), and sunflower honey (23 samples) were analyzed. 

It was observed that the mean values for most physicochemical parameters (glucose, fructose, sucrose content, 5-HMF levels, acidity, and electrical conductivity) varied significantly among different types of honey, years, and regions. Therefore, the statistics provided good differentiation between the honey samples in terms of their botanical origin, geographical origin, and year of harvest.

In a comparison of the measured parameters with the acceptable ranges defined by regulatory bodies, the quality and authenticity of the honey samples were assessed. Adulteration was identified in 22 honey samples, based on the higher sucrose, 5-HMF content, and acidity and low diastase activity, which did not meet recommended values for good honey quality. Additionally, 64 samples were non-compliant.

Nevertheless, a considerable number of honey samples from the Republic of Serbia could be deemed to have good honey quality. We showed that physicochemical parameters can serve as a good first step for the assessment of adulteration. These findings could be important for honey quality assurance, beekeeping practices, and identifying potential new approaches for further research.

In addition, the relationship between physicochemical data and the differentiation of honey types was investigated using a cANN. These findings demonstrate the reliability of the cANN model in accurately identifying acacia honey, honeydew honey, linden honey, monofloral honey, polyfloral honey, and sunflower honey based on physicochemical data analysis. Increasing the number of samples in the investigation may enhance the cANN model’s ability to discern subtle patterns specific to each type of honey.

## Figures and Tables

**Figure 1 foods-13-01530-f001:**
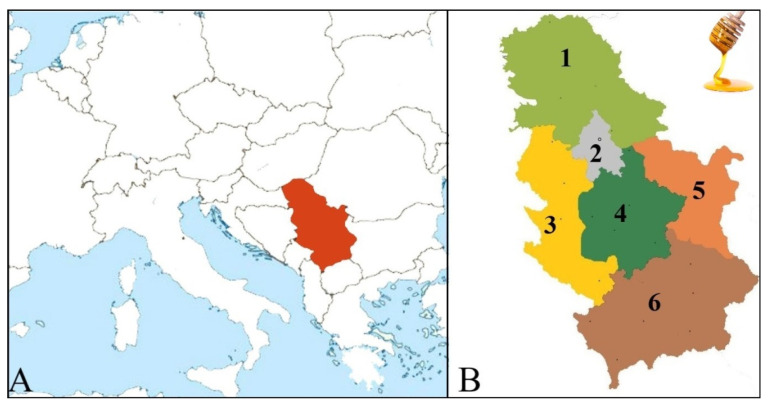
(**A**) Regional map of the Republic of Serbia in Europe; (**B**) map of the Republic of Serbia divided into six regions: 1. Northern region, 2. Belgrade, 3. Western region, 4. Central region, 5. Eastern region, and 6. Southern region.

**Table 1 foods-13-01530-t001:** Descriptive statistics of physicochemical parameters in 609 analyzed Serbian honey samples of different botanical origins.

Parameter	Type of Honey	N	Means ± SD	C. I. −95%	C. I. +95%	Var	Min	Max	Q25	Median	Q75
Glu	Acacia	213	27.40 ± 3.87 ^a^	26.88	27.93	14.96	16.02	40.00	24.83	26.40	29.70
	Honeydew	29	29.33 ± 5.17 ^ab^	27.37	31.30	26.69	16.19	38.85	26.69	29.92	33.10
	Linden	34	29.20 ± 3.91 ^ab^	27.83	30.56	15.25	17.18	35.74	27.11	29.14	31.00
	Monofloral	8	31.67 ± 7.53 ^abc^	25.37	37.96	56.66	13.44	36.10	32.22	34.07	35.61
	Polyfloral	302	31.39 ± 4.60 ^b^	30.87	31.91	21.14	15.72	44.80	28.95	31.57	34.40
	Sunflower	23	34.57 ± 3.98 ^c^	32.85	36.29	15.86	24.10	40.42	31.54	34.20	37.90
Fru	Acacia	213	40.83 ± 5.83 ^c^	40.04	41.62	34.01	13.60	53.10	37.61	41.35	44.99
	Honeydew	29	37.84 ± 6.32 ^abc^	35.44	40.25	39.96	22.65	49.72	35.64	38.97	42.81
	Linden	34	35.39 ± 5.82 ^a^	33.36	37.42	33.88	21.80	44.80	32.04	35.07	40.32
	Monofloral	8	35.41 ± 5.17 ^abc^	31.09	39.73	26.70	26.66	42.09	32.18	35.87	39.22
	Polyfloral	302	38.68 ± 5.20 ^b^	38.09	39.26	27.02	18.94	53.64	36.40	39.24	41.60
	Sunflower	23	37.83 ± 3.26 ^abc^	36.42	39.24	10.60	30.61	45.34	36.33	37.18	39.56
Suc	Acacia	213	1.19 ± 1.79 ^b^	0.94	1.43	3.21	<0.50	**16.58**	<0.50	0.57	1.34
	Honeydew	29	0.46 ± 0.61 ^ab^	0.23	0.69	0.37	<0.50	2.70	<0.50	<0.50	<0.50
	Linden	34	0.64 ± 1.08 ^ab^	0.27	1.02	1.16	<0.50	**5.60**	<0.50	<0.50	<0.50
	Monofloral	8	<0.50	/	/	/	<0.50	<0.50	<0.50	<0.50	<0.50
	Polyfloral	302	0.63 ± 1.41 ^a^	0.47	0.79	2.00	<0.50	**16.34**	<0.50	<0.50	<0.50
	Sunflower	23	0.34 ± 0.23 ^ab^	0.24	0.44	0.05	<0.50	1.03	<0.50	<0.50	<0.50
5-HMF	Acacia	213	5.71 ± 8.20 ^a^	4.60	6.82	67.25	<0.50	**54.80**	1.50	2.70	5.80
	Honeydew	29	6.34 ± 12.66 ^a^	1.52	11.15	160.27	<0.50	**63.41**	1.50	2.60	5.50
	Linden	34	5.41 ± 6.37 ^a^	3.18	7.63	40.63	<0.50	21.80	0.70	2.55	6.21
	Monofloral	8	11.84 ± 19.74 ^a^	−4.66	28.34	389.50	<0.50	**59.20**	1.03	5.00	11.60
	Polyfloral	302	8.31 ± 12.36 ^a^	6.91	9.71	152.84	<0.50	**93.50**	1.90	3.38	8.40
	Sunflower	23	4.85 ± 8.34 ^a^	1.25	8.46	69.56	<0.50	40.80	1.20	2.40	5.00
MC	Acacia	213	16.19 ± 1.07 ^a^	16.04	16.33	1.15	13.50	19.80	15.50	16.10	16.80
	Honeydew	29	16.36 ± 2.10 ^a^	15.56	17.15	4.40	13.40	**26.00**	15.40	16.40	16.80
	Linden	34	16.61 ± 1.10 ^ab^	16.23	17.00	1.21	14.20	19.80	16.00	16.60	17.10
	Monofloral	8	17.20 ± 2.04 ^ab^	15.49	18.91	4.16	15.40	**22.00**	16.20	16.70	17.20
	Polyfloral	302	16.45 ± 1.13 ^a^	16.33	16.58	1.28	13.00	19.90	15.70	16.45	17.10
	Sunflower	23	17.51 ± 1.37 ^b^	16.92	18.11	1.88	14.90	19.90	16.70	17.20	18.60
Acid	Acacia	213	11.44 ± 5.05 ^a^	10.76	12.12	25.48	2.30	31.43	8.00	9.77	12.00
	Honeydew	29	30.17 ± 8.07 ^d^	27.10	33.24	65.14	6.00	46.00	28.50	32.50	35.00
	Linden	34	16.33 ± 6.90 ^b^	13.92	18.74	47.62	5.50	34.72	11.00	15.00	20.35
	Monofloral	8	15.08 ± 8.16 ^abc^	8.25	21.90	66.55	7.50	30.00	9.60	11.46	20.50
	Polyfloral	302	21.04 ± 9.33 ^c^	19.98	22.09	86.98	4.00	**61.26**	14.50	20.61	25.50
	Sunflower	23	23.00 ± 7.22 ^c^	19.88	26.12	52.08	8.61	37.50	16.63	22.50	27.50
Dia	Acacia	213	13.06 ± 7.64 ^a^	12.03	14.09	58.37	**3.90**	114.00	10.25	12.50	14.30
	Honeydew	29	12.80 ± 3.14 ^a^	11.60	13.99	9.89	8.40	19.80	10.40	12.20	15.10
	Linden	34	12.90 ± 2.36 ^a^	12.08	13.72	5.59	8.40	19.07	11.00	13.15	14.10
	Monofloral	8	11.90 ± 2.87 ^a^	9.50	14.30	8.23	9.70	18.18	9.95	10.96	12.75
	Polyfloral	302	12.90 ± 3.43 ^a^	12.51	13.29	11.76	**0.50**	32.54	10.50	12.90	14.60
	Sunflower	23	13.45 ± 2.50 ^a^	12.37	14.53	6.27	9.60	17.42	11.06	13.70	15.70
Ins	Acacia	213	0.01 ± 0.01 ^a^	0.01	0.01	0	0	0.07	0.01	0.01	0.01
	Honeydew	29	0.01 ± 0.00 ^a^	0.01	0.01	0	0	0.01	0.01	0.01	0.01
	Linden	34	0.01 ± 0.00 ^a^	0.01	0.01	0	0	0.01	0.01	0.01	0.01
	Monofloral	8	0.01 ± 0.00 ^a^	0.01	0.01	0	0.01	0.02	0.01	0.01	0.01
	Polyfloral	302	0.01 ± 0.01 ^a^	0.01	0.01	0	0	0.10	0.01	0.01	0.01
	Sunflower	23	0.01 ± 0.01 ^a^	0.01	0.02	0	0	0.04	0.01	0.01	0.01
Econd	Acacia	213	0.20 ± 0.10 ^a^	0.18	0.21	0.01	**0.03**	0.69	0.14	0.18	0.20
	Honeydew	29	1.14 ± 0.27 ^e^	1.03	1.24	0.07	0.82	1.80	0.89	1.06	1.33
	Linden	34	0.48 ± 0.19 ^d^	0.42	0.54	0.03	**0.04**	0.78	0.35	0.53	0.60
	Monofloral	8	0.25 ± 0.16 ^ab^	0.12	0.38	0.02	0.12	0.52	0.15	0.17	0.36
	Polyfloral	302	0.40 ± 0.15 ^c^	0.38	0.42	0.02	**0.03**	0.79	0.32	0.40	0.47
	Sunflower	23	0.39 ± 0.09 ^bcd^	0.36	0.43	0.01	0.15	0.54	0.35	0.38	0.46

^a–e^ Means in the same column with different superscripts are statistically different (*p* ≤ 0.05). C. I.—confidence interval, N—number of samples, Var—variance, Min—minimum, Max—maximum, Q25—25% quantile, Q75—75% quantile. Type—type of honey; Glu—glucose [g/100 g]; Fru—fructose [g/100 g]; Suc—sucrose [g/100 g]; 5-HMF—5-hydroxymethylfurfural [mg/kg]; MC—moisture content [%]; Acid—acidity [meq/kg]; Dia—diastase activity [DN]; Ins—insoluble matter [%]; Econd—electrical conductivity [mS/cm]. **Bold values** (max or min) indicate values that were not in line with international requirements [1,2].

**Table 2 foods-13-01530-t002:** Descriptive statistics of physicochemical parameters in 609 analyzed Serbian honey samples of different years.

Parameter	Year	N	Means ± SD	C.I. −95%	C.I. +95%	Var	Min	Max	Q25	Median	Q75
Glu	2018	209	28.84 ± 4.89 ^a^	28.17	29.51	23.94	13.44	42.71	25.92	29.03	31.86
	2019	80	31.51 ± 4.44 ^b^	30.52	32.49	19.68	22.05	40.42	29.09	32.08	33.92
	2020	78	28.86 ± 4.23 ^a^	27.91	29.82	17.86	17.42	38.06	25.83	27.85	32.30
	2021	108	30.97 ± 4.87 ^b^	30.04	31.90	23.72	18.50	44.80	27.42	30.98	34.75
	2022	113	30.29 ± 4.79 ^ab^	29.40	31.19	22.98	16.51	41.00	26.19	30.13	34.10
	2023	21	30.53 ± 4.39 ^ab^	28.53	32.52	19.24	23.61	38.11	27.81	29.40	34.64
Fru	2018	209	37.13 ± 6.02 ^a^	36.31	37.95	36.30	18.94	49.50	33.65	37.69	41.12
	2019	80	39.73 ± 4.58 ^b^	38.71	40.75	20.98	27.84	49.71	38.02	39.58	42.86
	2020	78	40.95 ± 5.15 ^b^	39.79	42.11	26.54	30.45	53.64	37.18	41.11	44.98
	2021	108	39.76 ± 5.53 ^b^	38.70	40.81	30.64	20.20	53.10	36.75	39.85	42.77
	2022	113	39.97 ± 5.38 ^b^	38.97	40.98	28.98	13.60	50.69	37.58	40.18	43.40
	2023	21	42.25 ± 2.52 ^b^	41.10	43.39	6.37	38.27	47.17	41.12	42.18	43.56
Suc	2018	209	0.78 ± 1.67 ^a^	0.55	1.01	2.78	<0.50	**16.34**	<0.50	<0.50	<0.5
	2019	80	0.66 ± 0.69 ^a^	0.51	0.81	0.48	<0.50	3.05	<0.50	<0.50	0.78
	2020	78	0.96 ± 1.96 ^a^	0.52	1.40	3.83	<0.50	**16.58**	<0.50	<0.50	1.04
	2021	108	0.80 ± 1.65 ^a^	0.48	1.11	2.73	<0.50	**15.40**	<0.50	<0.50	0.75
	2022	113	0.68 ± 0.90 ^a^	0.51	0.85	0.81	<0.50	**5.60**	<0.50	<0.50	0.71
	2023	21	1.59 ± 1.90 ^a^	0.73	2.45	3.59	<0.50	6.48	<0.50	0.81	1.65
5-HMF	2018	209	9.92 ± 12.71 ^c^	8.19	11.65	161.49	<0.50	**63.41**	2.10	4.30	13.40
	2019	80	8.94 ± 14.61 ^bc^	5.69	12.19	213.41	<0.50	**93.50**	2.05	3.90	9.45
	2020	78	3.96 ± 4.28 ^a^	3.00	4.93	18.30	<0.50	22.10	1.20	2.65	4.50
	2021	108	4.56 ± 7.95 ^ab^	3.05	6.08	63.13	<0.50	**71.30**	1.55	2.45	4.30
	2022	113	5.44 ± 8.62 ^ab^	3.83	7.04	74.36	<0.50	**57.00**	0.70	2.40	5.60
	2023	21	4.49 ± 6.39 ^abc^	1.58	7.40	40.84	<0.50	30.00	1.50	2.60	5.60
MC	2018	209	16.34 ± 1.28 ^a^	16.17	16.52	1.64	13.40	**26.00**	15.60	16.40	17.00
	2019	80	16.29 ± 0.98 ^a^	16.07	16.50	0.96	14.70	19.10	15.50	16.25	16.90
	2020	78	16.46 ± 1.35 ^a^	16.16	16.76	1.82	13.00	19.90	15.50	16.30	17.20
	2021	108	16.52 ± 1.22 ^a^	16.29	16.76	1.48	13.70	19.90	15.75	16.40	17.20
	2022	113	16.49 ± 1.08 ^a^	16.29	16.69	1.16	13.90	19.90	15.80	16.60	17.10
	2023	21	16.50 ± 1.70 ^a^	15.73	17.28	2.89	13.10	**22.00**	15.70	16.50	17.00
Acid	2018	209	17.18 ± 8.71 ^ab^	15.99	18.36	75.78	2.30	44.50	9.50	15.00	24.00
	2019	80	20.21 ± 9.72 ^b^	18.04	22.37	94.51	6.00	46.00	10.00	20.25	27.50
	2020	78	17.36 ± 9.87 ^ab^	15.13	19.59	97.49	4.00	50.00	10.00	14.00	23.45
	2021	108	19.34 ± 7.61 ^b^	17.89	20.79	57.96	6.60	40.22	12.25	19.12	24.00
	2022	113	15.65 ± 7.84 ^a^	14.19	17.11	61.46	5.50	47.47	9.00	13.97	20.96
	2023	21	21.53 ± 20.14 ^ab^	12.36	30.70	405.75	8.10	**61.26**	9.92	16.00	21.95
Dia	2018	209	13.53 ± 7.99 ^a^	12.44	14.62	63.82	**0.50**	114.00	10.56	12.76	15.10
	2019	80	13.73 ± 3.11 ^a^	13.04	14.42	9.68	8.50	21.40	10.88	13.50	16.42
	2020	78	13.99 ± 1.84 ^a^	13.58	14.41	3.39	8.70	17.30	13.30	14.10	15.10
	2021	108	12.02 ± 2.39 ^a^	11.57	12.48	5.70	8.10	17.40	9.95	11.95	13.90
	2022	113	11.91 ± 3.17 ^a^	11.32	12.50	10.07	**3.90**	30.86	10.10	11.90	13.20
	2023	21	10.98 ± 1.61 ^a^	10.25	11.72	2.60	8.82	14.08	9.90	10.70	12.02
Ins	2018	209	0.01 ± 0.00 ^a^	0.01	0.01	0	0	0.02	0.01	0.01	0.01
	2019	80	0.01 ± 0.00 ^a^	0.01	0.01	0	0	0.02	0.01	0.01	0.01
	2020	78	0.01 ± 0.00 ^a^	0.01	0.01	0	0.01	0.02	0.01	0.01	0.01
	2021	108	0.01 ± 0.00 ^a^	0.01	0.01	0	0	0.01	0.01	0.01	0.01
	2022	113	0.01 ± 0.01 ^b^	0.01	0.02	0	0	0.10	0.01	0.01	0.01
	2023	21	0.02 ± 0.01 ^c^	0.01	0.02	0	0	0.04	0.01	0.02	0.02
Econd	2018	209	0.37 ± 0.24 ^ab^	0.34	0.40	0.06	**0.03**	1.56	0.19	0.32	0.47
	2019	80	0.46 ± 0.35 ^b^	0.38	0.54	0.12	**0.08**	1.80	0.20	0.38	0.55
	2020	78	0.31 ± 0.14 ^a^	0.27	0.34	0.02	0.10	0.75	0.19	0.33	0.38
	2021	108	0.39 ± 0.25 ^ab^	0.34	0.43	0.06	0.10	1.50	0.20	0.36	0.45
	2022	113	0.33 ± 0.20 ^a^	0.29	0.37	0.04	0.11	1.16	0.15	0.32	0.44
	2023	21	0.31 ± 0.16 ^ab^	0.24	0.38	0.03	0.13	0.63	0.14	0.34	0.45

^a–c^ Means in the same column with different superscripts are statistically different (*p* ≤ 0.05). C. I.—confidence interval, N—number of samples, Var—variance, Min—minimum, Max—maximum, Q25—25% quantile, Q75—75% quantile. Year—year of harvest; Glu—glucose [g/100 g]; Fru—fructose [g/100 g]; Suc—sucrose [g/100 g]; 5-HMF—5-hydroxymethylfurfural [mg/kg]; MC—moisture content [%]; Acid—acidity [meq/kg]; Dia—diastase activity [DN]; Ins—insoluble matter [%]; Econd—electrical conductivity [mS/cm]. **Bold values** (max or min) indicate values that were not in line with international requirements [1,2].

**Table 3 foods-13-01530-t003:** Descriptive statistics of physicochemical parameters in 609 analyzed Serbian honey samples from different regions.

Parameter	Region	N	Means ± SD	C.L. −95%	C.I. +95%	Var	Min	Max	Q25	Median	Q75
Glu	Western region	114	29.59 ± 5.34 ^a^	28.60	30.58	28.50	13.44	40.42	25.99	29.02	33.98
	Belgrade	205	30.30 ± 4.73 ^a^	29.65	30.95	22.34	16.02	42.71	27.13	30.82	33.12
	Northern region	149	30.02 ± 4.99 ^a^	29.21	30.83	24.93	17.18	44.80	26.34	29.70	33.48
	Central region	29	27.84 ± 4.03 ^a^	26.30	29.37	16.28	18.50	35.19	25.03	28.32	30.76
	Eastern region	92	29.72 ± 4.17 ^a^	28.85	30.58	17.40	17.42	39.57	26.88	29.78	33.18
	Southern region	20	30.56 ± 4.55 ^a^	28.43	32.69	20.70	23.85	37.31	25.59	32.06	34.29
Fru	Western region	114	38.93 ± 5.42 ^a^	37.93	39.94	29.38	13.60	51.04	36.50	39.36	42.49
	Belgrade	205	39.46 ± 5.43 ^a^	38.71	40.21	29.52	22.65	53.64	36.64	39.82	42.97
	Northern region	149	38.49 ± 6.26 ^a^	37.48	39.51	39.16	18.94	50.69	35.80	39.11	42.50
	Central region	29	38.34 ± 7.20 ^a^	35.60	41.08	51.79	24.67	53.10	33.72	38.63	44.68
	Eastern region	92	39.86 ± 4.25 ^a^	38.98	40.74	18.08	29.83	50.08	37.13	39.99	42.75
	Southern region	20	39.41 ± 7.12 ^a^	36.07	42.74	50.73	21.42	49.72	35.10	39.59	44.65
Suc	Western region	114	1.16 ± 2.16 ^b^	0.76	1.57	4.66	<0.50	**16.34**	<0.50	<0.50	1.04
	Belgrade	205	0.75 ± 1.35 ^ab^	0.56	0.93	1.83	<0.50	**15.40**	<0.50	<0.50	0.72
	Northern region	149	0.62 ± 0.97 ^a^	0.47	0.78	0.93	<0.50	**6.48**	<0.50	<0.50	0.58
	Central region	29	0.58 ± 0.71 ^ab^	0.31	0.85	0.51	<0.50	3.47	<0.50	<0.50	0.59
	Eastern region	92	0.85 ± 1.82 ^ab^	0.47	1.23	3.30	<0.50	**16.58**	<0.50	<0.50	0.92
	Southern region	20	0.63 ± 0.62 ^ab^	0.34	0.92	0.38	<0.50	2.43	<0.50	<0.50	0.90
5-HMF	Western region	114	5.62 ± 8.55 ^a^	4.04	7.21	73.02	<0.50	**58.60**	1.50	2.60	5.40
	Belgrade	205	8.01 ± 12.32 ^a^	6.32	9.71	151.88	<0.50	**93.50**	1.70	3.74	9.00
	Northern region	149	7.06 ± 10.44 ^a^	5.37	8.75	108.97	<0.50	**63.20**	1.50	3.00	7.20
	Central region	29	6.00 ± 8.53 ^a^	2.76	9.25	72.78	<0.50	37.40	2.40	2.80	5.30
	Eastern region	92	7.85 ± 12.02 ^a^	5.36	10.34	144.46	<0.50	**63.41**	1.80	3.00	6.30
	Southern region	20	3.35 ± 3.80 ^a^	1.58	5.13	14.43	0.80	16.80	1.25	2.00	3.20
MC	Western region	114	16.41 ± 1.15 ^ab^	16.20	16.62	1.33	13.00	19.90	15.60	16.40	17.10
	Belgrade	205	16.31 ± 1.17 ^a^	16.15	16.47	1.36	13.40	19.90	15.60	16.30	17.00
	Northern region	149	16.71 ± 1.19 ^b^	16.52	16.90	1.42	13.10	**22.00**	16.00	16.60	17.20
	Central region	29	16.31 ± 0.94 ^ab^	15.95	16.67	0.89	15.00	18.60	15.60	16.40	16.90
	Eastern region	92	16.31 ± 1.51 ^ab^	15.99	16.62	2.28	14.00	**26.00**	15.40	16.35	16.90
	Southern region	20	16.00 ± 0.97 ^ab^	15.55	16.45	0.94	13.90	17.80	15.45	16.00	16.65
Acid	Western region	114	18.11 ± 8.89 ^a^	16.46	19.76	79.07	5.50	50.00	11.00	16.00	23.28
	Belgrade	205	17.72 ± 9.12 ^a^	16.46	18.97	83.10	4.00	46.00	9.50	15.97	24.00
	Northern region	149	17.96 ± 8.41 ^a^	16.60	19.32	70.69	6.00	44.50	10.50	16.50	24.00
	Central region	29	15.87 ± 8.15 ^a^	12.77	18.97	66.47	6.97	31.00	9.00	12.00	21.50
	Eastern region	92	18.25 ± 12.46 ^a^	15.66	20.83	155.33	2.30	**61.26**	9.21	16.00	24.25
	Southern region	20	17.89 ± 7.54 ^a^	14.36	21.41	56.79	6.50	29.00	10.50	20.00	24.00
Dia	Western region	114	13.23 ± 3.72 ^a^	12.54	13.92	13.82	8.10	32.54	11.00	12.90	14.15
	Belgrade	205	13.28 ± 7.70 ^a^	12.22	14.34	59.30	**0.50**	114.00	10.30	12.70	14.80
	Northern region	149	12.94 ± 3.37 ^a^	12.39	13.48	11.39	8.05	31.69	10.40	12.97	14.50
	Central region	29	12.33 ± 2.92 ^a^	11.22	13.44	8.54	**3.90**	16.40	10.10	12.60	14.74
	Eastern region	92	12.15 ± 2.86 ^a^	11.55	12.74	8.19	8.20	23.50	9.70	11.80	13.83
	Southern region	20	12.98 ± 2.90 ^a^	11.62	14.34	8.42	8.40	17.05	10.65	12.54	15.90
Ins	Western region	114	0.01 ± 0.01 ^a^	0.01	0.01	0	0	0.04	0.01	0.01	0.01
	Belgrade	205	0.01 ± 0.00 ^a^	0.01	0.01	0	0	0.04	0.01	0.01	0.01
	Northern region	149	0.01 ± 0.01 ^a^	0.01	0.01	0	0	0.10	0.01	0.01	0.01
	Central region	29	0.01 ± 0.01 ^a^	0.01	0.02	0	0	0.07	0.01	0.01	0.01
	Eastern region	92	0.01 ± 0.00 ^a^	0.01	0.01	0	0	0.02	0.01	0.01	0.01
	Southern region	20	0.01 ± 0.00 ^a^	0.01	0.01	0	0	0.02	0.01	0.01	0.01
Econd	Western region	114	0.32 ± 0.19 ^a^	0.29	0.36	0.04	0.10	1.48	0.18	0.33	0.42
	Belgrade	205	0.38 ± 0.26 ^a^	0.34	0.41	0.07	**0.08**	1.80	0.20	0.34	0.44
	Northern region	149	0.36 ± 0.23 ^a^	0.33	0.40	0.05	**0.05**	1.54	0.19	0.35	0.45
	Central region	29	0.37 ± 0.29 ^a^	0.26	0.48	0.08	0.11	1.50	0.15	0.38	0.47
	Eastern region	92	0.39 ± 0.24 ^a^	0.34	0.44	0.06	**0.03**	1.19	0.19	0.37	0.53
	Southern region	20	0.44 ± 0.35 ^a^	0.27	0.60	0.12	0.11	1.50	0.17	0.31	0.59

^a,b^ Means in the same column with different superscripts are statistically different (*p* ≤ 0.05). C. I.—confidence interval, N—number of samples, Var—variance, Min—minimum, Max—maximum, Q25—25% quantile, Q75—75% quantile. Glu—glucose [g/100 g]; Fru—fructose [g/100 g]; Suc—sucrose [g/100 g]; 5-HMF—5-hydroxymethylfurfural [mg/kg]; MC—moisture content [%]; Acid—acidity [meq/kg]; Dia—diastase activity [DN]; Ins—insoluble matter [%]; Econd—electrical conductivity [mS/cm]. **Bold values** (max or min) indicate values that were not in line with international requirements [1,2].

**Table 4 foods-13-01530-t004:** Correlation matrix of physicochemical parameters in 609 analyzed Serbian honey samples.

	Fru	Suc	5HMF	MC	Acid	Dia	Ins	Econd
Glu	0.237 ***	−0.129 **	0.096 *	0.046	0.189 ***	0.004	−0.007	0.144 ***
Fru		−0.057	−0.087 *	−0.081 *	−0.076	−0.047	0.074	−0.105 *
Suc			0.001	−0.044	−0.150 ***	−0.023	−0.066	−0.144 ***
5HMF				−0.085 *	0.107 *	−0.129 **	−0.032	0.011
MC					0.147 ***	0.027	0.017	0.061
Acid						0.034	0.028	0.570 ***
Dia							−0.034	0.022
Ins								−0.023

*** Correlations significant at *p* < 0.001 level; ** Correlations significant at *p* < 0.01 level; * Correlations significant at *p* < 0.05 level; Glu—glucose [g/100 g]; Fru—fructose [g/100 g]; Suc—sucrose [g/100 g]; 5HMF—5-hydroxymethylfurfural [mg/kg]; MC—moisture content [%]; Acid—acidity [meq/kg]; Dia—diastase activity [DN]; Ins—insoluble matter [%]; Econd—electrical conductivity [mS/cm].

**Table 5 foods-13-01530-t005:** Artificial neural network model summary of training and testing cycles.

Network	Performance		Training Algorithm	Error Function	Activation	
	Train.	Test			Hidden	Output
MLP 8-12-6	89.98%	81.29%	BFGS 203	Entropy	Logistic	Softmax

MLP—multi-layer perceptron.

## Data Availability

The original contributions presented in the study are included in the article; further inquiries can be directed to the corresponding author.

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
