# Peer review of "Assessing the Impact of Botanical Origins, Harvest Years, and Geographical Variability on the Physicochemical Quality of Serbian Honey"

_foods, 2024, doi:10.3390/foods13101530_

Round 1
Reviewer 1 Report
Comments and Suggestions for Authors
In this study, the quality of 609 honey samples collected in the Republic of Serbia was investigated through physicochemical parameters such as electrical conductivity, sugar content (glucose and fructose as well as sucrose content), 5-HMF content, moisture content, acidity, diastase activity and insoluble matter. Study findings show how soil composition, altitude, geographic differences in climate, and regional factors change honey's distinctive characteristics and chemical profile.
Meanwhile, samples, in such studies, collected in a narrow area over a period of 1 or 2 years are generally studied. In this study, many samples collected from different regions over a period of 6 years were examined. The number of samples, year, and region factors provide important data in terms of the quality and composition of honey.
Compared with other published material, this study may shed light on new developments in regional beekeeping development and international competitiveness for Serbia. Additionally, this study design (year, number of samples, honey type, etc.) can be an example for studies in other regions.
The planning of the study, the number of samples, and the analyzes were planned very well. All results were statistically supported. References are sufficient and appropriate.
Author Response
Answers to Reviewers
All changes have been made to the Manuscript (in red), and all answers to referee comments are given here in red.
Reviewer 1
In this study, the quality of 609 honey samples collected in the Republic of Serbia was investigated through physicochemical parameters such as electrical conductivity, sugar content (glucose and fructose as well as sucrose content), 5-HMF content, moisture content, acidity, diastase activity and insoluble matter. Study findings show how soil composition, altitude, geographic differences in climate, and regional factors change honey's distinctive characteristics and chemical profile.
Meanwhile, samples, in such studies, collected in a narrow area over a period of 1 or 2 years are generally studied. In this study, many samples collected from different regions over a period of 6 years were examined. The number of samples, year, and region factors provide important data in terms of the quality and composition of honey.
Compared with other published material, this study may shed light on new developments in regional beekeeping development and international competitiveness for Serbia. Additionally, this study design (year, number of samples, honey type, etc.) can be an example for studies in other regions.
The planning of the study, the number of samples, and the analyzes were planned very well. All results were statistically supported. References are sufficient and appropriate.
AUTHORS: Thank you for your constructive approach to our investigation. We corrected the manuscript according to the comments of other reviewers, but we did not change the essence of the paper, which you have well noticed.
Reviewer 2 Report
Comments and Suggestions for Authors
Please correct the attached file

Need minor correction
Author Response
Answers to Reviewers
All changes have been made to the Manuscript (in red), and all answers to referee comments are given here in red.
Reviewer 2
Need minor correction.
AUTHORS: We appreciate the time to review our work and the corrections you have found. We corrected them directly in the manuscript, marked in red.
Reviewer 3 Report
Comments and Suggestions for Authors
Letter to Authors:
Dear Authors,
Thank you for submitting your paper titled "The Influence of Honey Types, Harvest Year, and Geographical Regions on the Quality Parameters of Honey Produced in the Republic of Serbia" by Tasić et al., to the Foods Journal. Your in-depth investigation on the chemical and physical characterization of Serbian honey is of significant interest in the field if Food Science and Technology. However, after conducting a thorough review on your paper, I have determined that it requires major revisions before being considered for publication. Please carefully address the following issues:
[1] Please consider revising the title to: "Assessing the Impact of Botanical Origins, Harvest Years, and Geographical Variability on the Physicochemical Quality of Serbian Honey".
[2] Abstract (Lines 17-30): The authors mentioned that 22 honey samples could be considered adulterated based on various chemical and physical features, but they did not specify reference standards/criteria adopted for this determination.
[3] Introduction (Lines 34-45): Please highlight clearly the objectives and the gaps that need to be filled.
[4] Materials and Methods (Lines 84-92): The authors did not mention how the samples were randomized or if any stratification was adopted to ensure representativeness of the regions.
[5] Lines (144-152): Please justify your use of particular statistical methods to analyze your data.
[6] Lines (154-232): Please discuss the chemical and physical properties found with those in the literature to contextualize your findings.
[7] Results and Discussion (Lines 371-502): The authors extensively described their results but omitted to draw clear conclusions and the implications of their findings in the field.
[8] Tables 1, 2, and 3 are heavily packed with data and suffer from poor descriptions and discussion throughout the text. Please revise this point to ensure clarity and comprehensibility.
[9] Conclusions (Lines 560-583): The conclusion is overly vague. The authors need to summarize their findings clearly and concisely, and suggest practical applications and directions for future research.
[10] References Section: Please update the references section with more recent studies and not just outdated ones.
Comments on the Quality of English LanguageThe paper requires moderate English editing to enhance clarity and comprehensibility.
Author Response
Answers to Reviewers
All changes have been made to the Manuscript (in red), and all answers to referee comments are given here in red.
Reviewer 3
Dear Authors,
Thank you for submitting your paper titled "The Influence of Honey Types, Harvest Year, and Geographical Regions on the Quality Parameters of Honey Produced in the Republic of Serbia" by Tasić et al., to the Foods Journal. Your in-depth investigation on the chemical and physical characterization of Serbian honey is of significant interest in the field if Food Science and Technology.
AUTHORS: Thank you for your positive attitude toward our investigation. We corrected the Manuscript, according to the Reviewers’ comments, with the red text directly in the Manuscript.
However, after conducting a thorough review on your paper, I have determined that it requires major revisions before being considered for publication. Please carefully address the following issues:
AUTHORS: Thank you for your valuable comments.
[1] Please consider revising the title to: "Assessing the Impact of Botanical Origins, Harvest Years, and Geographical Variability on the Physicochemical Quality of Serbian Honey".
AUTHORS: The title of the Manuscript was changed according to the Reviewer’s suggestion.
[2] Abstract (Lines 17-30): The authors mentioned that 22 honey samples could be considered adulterated based on various chemical and physical features, but they did not specify reference standards/criteria adopted for this determination.
AUTHORS: Thank you for your observation. Indeed, the clarification regarding the reference standards or criteria used to determine adulteration in the honey samples is pivotal for the comprehensibility and reliability of the study. However, the abstract provided a summary of the findings, while the detailed methodology section in the full text of the paper gives the specific reference standards and criteria employed for identifying adulterated honey samples. We acknowledge the importance of explicitly mentioning these details within the main text.
[3] Introduction (Lines 34-45): Please highlight clearly the objectives and the gaps that need to be filled.
AUTHORS: Thanks for the feedback. We appreciate your suggestion to highlight the objectives and the gaps, but we would prefer to do this at the end of the section Introduction rather than at the beginning. We will make the necessary adjustments to ensure that the goals and identified gaps are clearly stated at the end of the Introduction.
[4] Materials and Methods (Lines 84-92): The authors did not mention how the samples were randomized or if any stratification was adopted to ensure representativeness of the regions.
AUTHORS: Although randomization of regions was not conducted initially, the study ensured the representativeness of the samples through careful consideration of their characteristics, including honey type, geographical origin, and year of collection. The samples were obtained directly from local beekeepers, and while randomization wasn't implemented at the outset, the collected samples were objectively characterized. Subsequently, the samples were analyzed, and statistical processing was applied to classify them based on the type of honey, region, and year of collection. This process helped to ensure that the samples represented the diversity of beekeeping regions under this study.
[5] Lines (144-152): Please justify your use of particular statistical methods to analyze your data.
AUTHORS: As mentioned in the text, the experimental results of statistical analysis for honey sample physicochemical parameters (glucose, fructose, sucrose content, 5-HMF levels, acidity, and electrical conductivity) were presented as the mean ± standard deviation for all parameters across the samples. These basic statistical analyses present the main trends of the collected data. Furthermore, Tukey’s HSD test was employed to test the differences between the mean values of honey samples (categorical variables were: the botanical origin of samples, year of production, and the region). All data underwent statistical processing (including descriptive statistics and Pearson’s correlation analysis) using the software package STATISTICA 10.0 (StatSoft Inc., Tulsa, OK, USA).
[6] Lines (154-232): Please discuss the chemical and physical properties found with those in the literature to contextualize your findings.
AUTHORS: Thank you. We have discussed some properties with those in the literature. We have added sentences that complement the previous text. We hope that we complete and contextualize our findings in the way the Reviewer intended.
[7] Results and Discussion (Lines 371-502): The authors extensively described their results but omitted to draw clear conclusions and the implications of their findings in the field.
AUTHORS: Thank you very much for this comment. We have added a few lines of text throughout the manuscript to better explain the statistically obtained results.
[8] Tables 1, 2, and 3 are heavily packed with data and suffer from poor descriptions and discussion throughout the text. Please revise this point to ensure clarity and comprehensibility.
AUTHORS: We have improved the text.
[9] Conclusions (Lines 560-583): The conclusion is overly vague. The authors need to summarize their findings clearly and concisely, and suggest practical applications and directions for future research.
AUTHORS: We changed the Conclusion section, according to the Reviewer’s comment. As mentioned in the article, these results could be important for honey quality assurance, and beekeeping practice, as well as for identifying potential new approaches for further research.
[10] References Section: Please update the references section with more recent studies and not just outdated ones.
AUTHORS: We added some references from 2023 and have improved the text.
Reviewer 4 Report
Comments and Suggestions for Authors
The reviewed article is well-written and the data presented is correctly described and discussed in the article. Nevertheless, a few minor corrections could improve the quality of the article and make the data easier for readers to perceive. Below are my detailed comments:
1. Please write clearly in section 2.1 what was the distribution of samples: how many from which region, year and type.
2. In section 2.3.1, please provide the exact procedure for analyzing diastase activity or add a citation
3. Please describe in more detail how honey samples were prepared for 5-HMF determination by chromatography
4. In line 157 and 158, the table descriptions are swapped in place.
5. In line 160, the authors report that some results are not in accordance with international requirements. Perhaps it would be possible to somehow indicate in Tables 1-3 which samples and parameters are in question?
6. From the data obtained, 5 honey samples were found to be adulterated on the basis of sucrose content, 12 on the basis of 5-HMF content and 4 on the basis of diastase activity. Did any of the samples perform badly in more than one test?
Author Response
Answers to Reviewers
All changes have been made to the Manuscript (in red), and all answers to referee comments are given here in red.
Reviewer 4
The reviewed article is well-written and the data presented is correctly described and discussed in the article. Nevertheless, a few minor corrections could improve the quality of the article and make the data easier for readers to perceive. Below are my detailed comments:
AUTHORS: Thank you for your positive attitude toward our investigation. We corrected the Manuscript, according to the Reviewers’ comments, with the red text directly in the Manuscript.
- Please write clearly in section 2.1 what was the distribution of samples: how many from which region, year and type.
AUTHORS: Thank you for this suggestion. We added a few sentences in this section, according to this comment.
- In section 2.3.1, please provide the exact procedure for analyzing diastase activity or add a citation
AUTHORS: Thank you. We added a reference.
- Please describe in more detail how honey samples were prepared for 5-HMF determination by chromatography
AUTHORS: Thank you. We have added some detailed information.
- In line 157 and 158, the table descriptions are swapped in place.
AUTHORS: Thank you for this observation!
- In line 160, the authors report that some results are not in accordance with international requirements. Perhaps it would be possible to somehow indicate in Tables 1-3 which samples and parameters are in question?
AUTHORS: Thank you. We have assigned (in bold numbers) the results of minimum or maximum values in Tables 1-3 that do not follow international requirements.
- From the data obtained, 5 honey samples were found to be adulterated on the basis of sucrose content, 12 on the basis of 5-HMF content and 4 on the basis of diastase activity. Did any of the samples perform badly in more than one test?
AUTHORS: Thank you for asking. Of 22 samples that were marked as adulterated, in 18 samples was found only one deviation. Otherwise, among higher sucrose content than recommended values, in four samples was noted lower content of the sum of glucose and fructose. Those were acacia honey from the Eastern region harvested in 2020 (in which sum of glucose and fructose was 52.72 mg/kg), linden honey from the Western region harvested in 2022 (with 55.80 mg/kg), polyfloral honey from Belgrade harvested in 2021 (52.16 mg/kg), and polyfloral honey from the Western region harvested in 2018 (49.50 mg/kg). We have added section 3.2. with several sentences about it, as well as an explanation about which samples are adulterated.
Round 2
Reviewer 3 Report
Comments and Suggestions for Authors
Dear Authors,
I am satisfied with your revision; therefore, your paper can be published in its current form.
Congratulations!
Author Response
Dear Reviewer,
Thank you very much.
Kind regards,
Authors